# Preparation, Characterization and Antioxidant Activity of Glycosylated Whey Protein Isolate/Proanthocyanidin Compounds

**DOI:** 10.3390/foods12112153

**Published:** 2023-05-26

**Authors:** Yaochang Li, Lian Zhou, Haizhi Zhang, Gang Liu, Xinguang Qin

**Affiliations:** 1College of Food Science and Engineering, Wuhan Polytechnic University, Wuhan 430023, China; 2Key Laboratory for Deep Processing of Major Grain and Oil, Wuhan Polytechnic University, Ministry of Education, Wuhan 430023, China

**Keywords:** whey protein isolate, proanthocyanidins, Maillard reaction, glycosylation, antioxidation

## Abstract

A glycosylated protein/procyanidin complex was prepared by self-assembly of glycosylated whey protein isolate and proanthocyanidins (PCs). The complex was characterized through endogenous fluorescence spectroscopy, polyacrylamide gel electrophoresis, Fourier infrared spectroscopy, oil–water interfacial tension, and transmission electron microscopy. The results showed that the degree of protein aggregation could be regulated by controlling the added amount of procyanidin, and the main interaction force between glycosylated protein and PCs was hydrogen bonding or hydrophobic interaction. The optimal binding ratio of protein:PCs was 1:1 (*w*/*w*), and the solution pH was 6.0. The resulting glycosylated protein/PC compounds had a particle size of about 119 nm. They exhibited excellent antioxidant and free radical-scavenging abilities. Moreover, the thermal denaturation temperature rose to 113.33 °C. Confocal laser scanning microscopy (CLSM) images show that the emulsion maintains a thick interface layer and improves oxidation resistance with the addition of PCs, increasing the application potential in the functional food industry.

## 1. Introduction

Proanthocyanidins (PCs) are oligomeric and polymeric flavan-3-ols, which are secondary metabolites of plants [1,2,3]. They are mainly found in the peels, shells, and seeds of plants, such as *Lycium ruthenicum* Murr, grape seeds, and apple peels [4]. PCs have excellent biological activities, including anti-inflammatory, antioxidant, and antibacterial capabilities [5,6]. After being dissolved in water, PCs are easily oxidized by oxygen in the air under the action of PC oxidase. At this time, the phenolic hydroxyl group in PCs undergoes decomposition to generate oxygen anions. With the continued loss of hydrogen atoms during oxidation, *o*-quinones are formed, resulting in a deepening of the purple-red color in the aqueous solution of PCs. The hydroxyl component of the procyanidin structure can act as a hydrogen atom donor. It contains multiple electrons, and it can undergo conjugation with double bonds; thus, PCs acquire excellent free radical-scavenging and antioxidant capabilities [7,8]. Given these properties, PCs are widely used in the development of functional foods.

Whey protein isolate (WPI) is a special byproduct of cheese production. Filtered whey protein is rich in α-lactalbumin, β-lactoglobulin, bovine serum albumin (BSA), and immunoglobulin [9,10]. As an amphiphilic biological macromolecule, WPI possesses a large number of amino acids on its surface, charged hydrophilic and hydrophobic regions, and a complex spatial structure that provides the basis for its interaction with other substances. However, WPI has some defects, such as easy aggregation and limited antioxidant capability in an acid–thermal environment. A previous laboratory investigation [11,12] showed that glycosylated protein/polyphenol compounds prepared by combining glycosylated protein and polyphenols could effectively overcome these problems. Furthermore, numerous studies have been conducted on the interaction of proteins and carbohydrates or polyphenols to improve protein stability [13,14]. Li et al. [15] used an ultrasonic-assisted Maillard reaction to prepare myofibrillar protein and dextran conjugates. They also investigated the solubility and emulsification capability properties. The solubility, emulsification capability, and stability of myofibrillar protein were improved by the covalent interaction between myofibrillar protein and dextran that occurred in the later stage of the Maillard reaction. In addition to the interaction between proteins and carbohydrates, the interaction between proteins and polyphenols has attracted considerable attention [16]. Yan et al. [17] found that protein isolate and protein–phenolic complex exhibited superior essential amino acid composition and similar protein subunits. Furthermore, the protein–phenolic complex exhibited higher solubility, as well as better foaming and emulsifying properties compared to the protein isolate. Numerous studies have substantiated that polyphenols exhibit robust binding affinities toward proteins through noncovalent and/or covalent bonds. As a result, these interactions lead to structural and property modifications, consequently exerting a significant influence on the bioaccessibility and bioavailability of polyphenols [16]. As a result, the thermal stability, foaming properties, and emulsifying properties of the protein–phenolic complex could be improved and developed as new food materials.

Compared with studies exploring whether the combination of phenols and proteins could improve the functional properties of proteins, few studies on the interaction of WPI and glycosylated WPI (WPI-Lac) with PCs are available. Previous research results have demonstrated the stability of the WPI-Lac binary complex and that PCs, WPI, and WPI-Lac could further form three-phase protein–carbohydrate/phenol compounds. The present work aims to study the effect of PCs on the functional properties of proteins and glycosylated proteins; explore the interactions among PCs, WPI, and WPI-Lac and the antioxidant capacity of these compounds; and provide data to support the development of new antioxidant protein-based compounds and the establishment of a Pickering emulsion system with anti-lipid peroxidation properties.

## 2. Materials and Methods

### 2.1. Samples, Chemicals and Standard

The protein WPI #9420 was kindly provided by Hilmar Ingredients (Hilmar, CA, USA); the ingredients include α-whey protein, β-lactoglobulin, bovine serum protein (BSA), and immunoglobulin, mainly β-lactoglobulin. D-lactose for glycation was obtained from Tokyo Chemical Industry Co., Ltd. (Tokyo, Japan). PCs were purchased from Shanghai Yuanye Biotechnology Co., Ltd. (Shanghai, China) with the formula C_30_H_26_O_13_ (CAS#4852-22-6). All other chemicals were of analytical grade.

### 2.2. Preparation of Glycosylated Protein/PC Compounds

(1)Preparation of glycosylated protein: In the existing process [12], WPI and lactose were dissolved in deionized water with a mass ratio of 1:1 (*w*/*w*). After two hours of stirring, the mixture was kept at 4 °C overnight to fully hydrate the protein with a pH of 7 in preparation for freeze drying. The powdered WPI and lactose mixture was placed in a desiccator at the humidity of 79% and reacted at 65 °C for 24 h to obtain WPI-Lac. The WPI-Lac powder was dissolved in deionized water and dialyzed with a 3500 kDa dialysis bag for 72 h. Finally, the dialysate was removed from the dialysis bag, freeze-dried, and stored.(2)Preparation of glycosylated protein/PC compounds: WPI and WPI-Lac solutions were prepared, and the system was kept at pH 6 while adding PC solution. At 25 °C, the PC solution was slowly added to the protein solution under stirring for 2 h. The concentration of protein solution remained unchanged (0.5% by weight), but in the mixture, the mass ratio of protein to PCs changed.

### 2.3. Fluorescence Spectral Analysis

WPI, WPI-Lac, and PC solutions were prepared using deionized water. The protein/PC solutions were prepared with different mass ratios (1:0, 1:2, 1:4, 1:6, 1:8, and 1:10). The settings of the fluorescence spectrophotometer (FL-4600, Hitachi Instruments Engineering Co., Ltd., Tokyo, Japan) were as follows: excitation wavelength of 280 nm and emission wavelength range of 300–450 nm.

### 2.4. Particle Characteristics

A dynamic light scatterer (ZetasizerNano-ZS, Malvern Instruments Ltd., Worcestershire, UK) was used to determine the particle size, polydispersity index (PDI), and zeta potential of WPI or WPI-Lac and PCs. Among the measured samples, the refractive index of protein was 1.450, and that of water was 1.330.

### 2.5. Turbidity

Sample solutions of the complexes with different pH values were prepared. The transmittance T(%) of the compound was determined using an ultraviolet spectrophotometer (UV-1000, AOE Instrument Co., Ltd., Shanghai, China) at 600 nm. Turbidity was expressed as 100-T (%), with ultrapure water as a blank.

### 2.6. SDS-PAGE

WPI, WPI/PC, WPI-Lac, and WPI-Lac/PC solutions with concentrations of 2 mg/mL were prepared. A total of 40 μL of sample solution was added to 10 μL of loading buffer. The mixture was centrifuged at 3000 rpm for 3 min and soaked in boiling water for 15 min to completely denature it. SDS-PAGE electrophoresis was performed with a discontinuous buffer system comprising 5% concentrated gel and 12% separating gel with the loading volume of 7 μL. The bromophenol blue indicator was introduced into the separation gel; first, it was set to 60 V, and then it was adjusted to 120 V. The colloids were then stained with Coomassie blue to change their color until the streaks were clearly visible.

### 2.7. Fourier Transform Infrared Spectroscopic Analysis

A Fourier transform infrared spectrometer (NEXUS 670, Nicolet Co., Ltd., Middleton, MA, USA) was used to analyze the structural changes in the samples. First, lyophilized samples were ground and uniformly mixed with KBr at a weight ratio of 1:100 and then ground with potassium bromide to prepare potassium bromide tablets for analysis with the following parameter settings: wavenumber range of 4000−400 cm^−1^, resolution of 8 cm^−1^, and scan times of 16.

### 2.8. Determination of Surface Hydrophobicity

The protein solution was diluted to different concentrations with 10 mM phosphate buffer (pH 7.0). Then, 8 mmol/L ANS (dissolved in 0.01 mol/L buffer, pH 6.7) was added and mixed thoroughly. The solution was left at room temperature, protected from light, for 15 min. Fluorescence intensity was determined by fluorescence spectroscopy. The excitation wavelength was set to 390 nm, and the emission wavelength was set to 400–700 nm. On this basis, the fluorescence intensity was used to perform curve fitting on the protein concentration, and the obtained curve was H_0_.

### 2.9. Interfacial Tension Measurement

The oil–water interfacial tension of the sample droplets was measured using an interfacial rheometer (DSA30R, Kruss Company, Hamburg, Germany). The interfacial tension between WPI or WPI-Lac/PCs (5 mg/mL) and MCT was measured by the suspension drop method. Sample droplets were generated in a cuvette filled with MCT by a syringe (outer diameter: 1.81 mm), 25 uL of droplets were obtained at an output rate of 30 uL/s to avoid droplet desorption, and the interfacial tension was determined after 1 h. Interfacial tension was calculated by computer-automated rapid acquisition of droplet images, edge detection, and fitting of the Young–Laplace equation.

### 2.10. Differential Scanning Calorimetry

The samples were placed in a differential scanning calorimeter (Q2000, TA Instruments, New Castle, DE, USA) for DSC analysis according to a previous report [18] with some modifications. Samples totaling 5 ± 0.2 mg were weighed into aluminum containers and compressed to degas them. An aluminum box with a similar quality was used as a reference. The scanning temperature was 20–200 °C, the heating rate was 10 °C/min, and the nitrogen flow rate was 20 mL/min. The initial denaturation temperature (T_m_), denaturation peak temperature (T_p_), and denaturation entropy (ΔH) were calculated and analyzed using general 5.4 TA software.

### 2.11. Transmission Electron Microscopy (TEM)

TEM (Libra 200MC type, Germany) was used to analyze the morphology of the compounds. The sample solution was dropped onto the copper grid and allowed to absorb for at least 20 min. After drying, the samples were observed and imaged with a 100 kV electron microscope.

### 2.12. Analysis of Antioxidant Properties of Compounds

(1)Determination of ABTS free radical-scavenging rate

First, a solution of 7 mM ABTS and 4.9 mM Na_2_S_2_O_8_ was prepared and stirred with an equal volume of ABTS stock solution. Then it was placed in the dark for 12–16 h. Before use, the ABTS stock solution was diluted to 734 nm with absolute ethanol to obtain 0.700 + 0.020. This solution was the working solution of ABTS. Then, 100 microliters of protein solution was mixed with 3 mL of ABTS working solution and incubated at 30 °C for 10 min. Absorption at 734 nm was measured. The free radical-scavenging rate of ABTS was calculated with the following formula:$$\mathrm{Clearance}\;\mathrm{rate}\;\%=\frac{A_{\mathrm{control}}-A_{\mathrm{sample}}}{A_{\mathrm{control}}}\;\times \;100,$$
where *A_control_* represents the absorbance of the blank control, and absolute ethanol was used as the blank control in this experiment; *A_sample_* represents the absorbance of the sample.

(2)Determination of DPPH free radical-scavenging rate

A total of 1 mL of protein solution and 1 mL of 0.2 mM DPPH working solution were completely mixed and reacted in the dark at room temperature for 20 min. Next, absorbance was measured at 517 nm. The free radical-scavenging rate of DPPH was calculated as ABTS.

(3)Determination of total antioxidant capacity by FRAP method

The reducing power of the sample was determined using an oxidation resistance test kit (Beyotime Biotechnology). The working solution was freshly prepared by mixing the dilution solution, TPTZ, and assay buffer solution in a ratio of 10:1:1 (*v*/*v*/*v*). Then, 180 μL of FRAP working solution was mixed with 5 μL of diluted sample and incubated at 37 °C for 5 min. Determination of absorbance at 593 nm was performed with FeSO_4_•7H_2_O provided in the kit; the solution was used to obtain a standard curve.

### 2.13. Preparation of Pickering Emulsion Stabilized by Glycosylated Protein/PC Compounds

A sample solution of 5 mg/mL was prepared and then mixed with tea oil. The mass of this emulsion was 10 g, and the mass fraction of the oil phase was 50%. The Pickering emulsion was obtained by stirring at 10,000 rpm for 2 min with a high-speed homogenizer (IKAT18, IKA-Werke Co., Ltd., Staufen im Breisgau, Germany) and measured by laser particle sizer (Mastersizer 3000, Malvern Instruments Ltd., Westborough, MA, USA). Finally, the emulsion was observed using a confocal laser scanning microscope [19].

### 2.14. Statistical Analysis of Data

Three separate runs of the entire experiment were completed, and all data were evaluated using one-way ANOVA and Duncan’s test with SPSS 19.0. *p* < 0.05 was taken as a significant difference, and the findings are shown as mean ± standard deviation.

## 3. Results

### 3.1. Fluorescence Spectral Analysis

When stimulated at 280 nm, tryptophan (Trp), tyrosine, and phenylalanine show luminous absorption peaks [20]. WPI contains an abundance of Trp residues, and Trp is sensitive to changes in the environment and protein structure and is frequently used as a fluorescent probe to study the interaction between proteins and polyphenols. In this study, the endogenous fluorescence intensity of WPI and WPI-Lac was used to evaluate the change in protein structures after the noncovalent binding of proteins to PCs. Figure 1a,b show that at the excitation wavelength of 280 nm, the protein sample without PCs has a very significant fluorescence emission peak at 342 nm. The endogenous fluorescence intensity of WPI-Lac is lower than that of WPI for two reasons. Glycosylation alters the environment of the Trp group, resulting in a decrease in fluorescence intensity. This is comparable to the effects found in the research of Chen et al. [21] on the relationship between glycosylated proteins and tannins. Second, the other amino acids in WPI combined with lactose to form a copolymer molecular layer that reduced the fluorescence intensity of the proteins. Yu et al. [22] studied the binding of peanut protein and lactose and reached similar conclusions.

The fluorescence of WPI and WPI-Lac significantly decreased after the addition of PCs. Additionally, when PC concentration increased, the highest absorption peak of the material shifted 6 nm toward the blue, from around 342 nm to approximately 336 nm. This result suggests that PCs can change the spatial conformation of the protein and the microstructure near Trp, the environment changed from hydrophobic to hydrophilic, and the peptide chain stretched further [23]. As the amount of PCs was increased, the fluorescence intensity gradually decreased, indicating that the addition of PCs changed the Trp microenvironment on the protein side chain; this is also consistent with the results of Sun et al. [24] showing that changes in protein fluorescence intensity may be related to changes in the microenvironment of Trp as proanthocyanidin concentrations increase. The interaction between protein and polyphenols was similar to that found by Zou et al. [25] and Zhou et al. [26] for soy protein isolate and grape seed procyanidins or epigallocatechin gallate.

### 3.2. Preparation of Glycosylated Protein/PC Compounds

(1)Effect of PC addition on the degree of aggregation of WPI and WPI-Lac

Table 1 shows the change in particle size, PDI, and zeta potential of the complex composed of protein and PCs at pH 6. Without PCs, the particle sizes of WPI and WPI-Lac were 246.8 ± 19.52 (PDI = 0.522 ± 0.068) and 358.8 ± 13.2 nm (PDI = 0.573 ± 0.023), respectively. With the increase in PC addition amount, the particle size and PDI of WPI/PCs and WPI-Lac/PCs decreased. When the ratio of proteins to PCs was 1:1 (*w*/*w*), the particle size and PDI of WPI/PCs and WPI-Lac/PCs decreased to 112.6 ± 10.3 (PDI = 0.242 ± 0.039) and 119.4 ± 39.8 nm (PDI = 0.226 ± 0.045), respectively. Adding PCs to protein slightly increased its absolute zeta potential, as shown in Table 1. This phenomenon indicated that the addition of PCs affects the spatial structure of WPI and WPI-Lac compounds. This result was similar to the previous results obtained for the formation of compounds with epigallocatechin gallate [27] and chitooligosaccharides [28]. This change indicated that the compound particles showed increased stability and did not easily aggregate after the addition of PCs [29].

(2)Effect of PCs on the degree of aggregation of WPI and WPI-Lac at different pH levels

Table 2 shows the characteristics of protein/PC compounds at a mass ratio of 1:1 (*w*/*w*) at different pH values. At pH 2, particle size and PDI cannot be determined due to low complex solubility. When the pH was 4~5, the particle size of WPI/PCs and WPI-Lac/PCs increased significantly, which was due to the neutralization of the pH near the isoelectric point [27]. In the pH 6 environment, the particle size of WPI/PCs and WPI-Lac/PC compounds was the smallest, and the value of PDI was small. This situation indicated that the compound particles were evenly distributed. The variation in the zeta potential shows that the absolute zeta potential is lower than 30 mV at pH 4 and 5. In addition, in the case of pH 3–8, the absolute zeta potential exceeds 30 mV, indicating that the composite particles have good and strong stability.

### 3.3. Turbidity

At various pH values, the compounds’ turbidity was assessed. According to our earlier work, the protein concentration was 5 mg/mL and the protein-to-PC ratio was 1:1 (*w*/*w*). When WPI-Lac and WPI are compared in Figure 2, it is clear that the latter has reduced turbidity at the isoelectric point. Because the PC solution was opaque at pH values 2 and 3, the turbidity of the WPI/PC and WPI-Lac/PC compounds was 100% under these circumstances. Because of the strong ionic contact between the protein and the PCs, which caused the compounds to precipitate out of the solution, the turbidity of the compounds remained at 100% at pH values 4 and 5. Due to the binding of glycosylated modified proteins to PCs, which increases their solubility, the turbidity of WPI-Lac/PCs was slightly lower at pH 6 than that of WPI/PCs, at 58% compared to 62% [27]. Moreover, the gradual decrement in the turbidity of the compounds at pH values 6–8 indicated the interaction between proteins and PCs [30].

### 3.4. SDS-PAGE

Figure 3a shows SDS-PAGE results for WPI, WPI/PCs, WPI-Lac, and WPI-Lac/PCs. The electrophoresis strips were analyzed using Image Lab5.1 software, and the results are provided in Figure 3b and Table 3. Previous studies have reported that WPI mainly resolves into three main bands: α-lactalbumin (13.8 kDa), β-lactoglobulin (16.0 kDa), and BSA (70.0 kDa) [31]. Lane 1 was WPI and showed obvious characteristic bands at 12.1, 14.5, and 71.7 kDa, and the remaining bands could be attributed to other protein components. Lane 3 was WPI-Lac. After glycosylation, the Maillard reaction increased the molecular weights of lactalbumin, β-lactoglobulin, and BSA to 13.7, 16.7, and 83.7 kDa, respectively. Lanes 2 and 4 were the bands of WPI/PCs and WPI-Lac/PCs, respectively. Comparison of WPI with WPI/PCs and WPI-Lac/PCs showed similar binding to previously obtained proteins and polyphenols [32]; the electrophoresis bands became unclear and tended to shift upwards [33]. This also indicated that the affinity between PCs and protein is caused by weak hydrogen bonds and hydrophobic interactions.

### 3.5. FTIR Analysis

The FTIR spectra of WPI, WPI-Lac, WPI/PCs, WPI-Lac/PCs, and PCs are shown in Figure 4. By comparing the IR spectra of WPI and WPI-Lac, it was shown that the latter displays a notable absorption peak in the range of 1200~1000 cm^−1^, which is caused by the stretching vibration of the -OH and C-O groups, therefore proving their covalence. The infrared absorption peaks of WPI are 3299 cm^−1^ (amide A region, indicating -NH elongation and hydrogen bonding), 1640 cm^−1^ (indicating C=O stretching/hydrogen bonding and COO-), 1640 cm^−1^ (amide I band, indicating C-N stretching), 2961 cm^−1^ (amide B, indicating C-N-N stretching), 1547 cm^−1^ (amide II band, representing C-N stretching and N-H bending), and 1230 cm^−1^ (amide III strip, indicating C is C-N stretching and N-H bending) [34]. In addition, the broadening of the absorption peak of WPI-Lac at the amide A band of 3700–3200 cm^−1^ (representing N–H stretching and hydrogen bonding) proved that the number of hydroxyl groups increased after WPI covalently bound to lactose. The Maillard reaction is associated with the formation of amide bonds (C-O, N-H, and C-N), and changes in the two absorption peaks described above demonstrate the occurrence of a glycosylation reaction.

The absorption wavelengths of proanthocyanidins were 3384, 1610, 1518, 1441, 1362, 1203, 1141, 1111, 1063, and 825–736 cm^−1^. This absorption induces vibrations of the phenol hydroxyl O-H bond, C=aromatic ring, -CH stretching vibration (alkane), CO stretching vibration (alcohol), C-O-C stretching vibration (ether), and =CH out-of-plane bending (aromatic ring) [35].

The distinctive absorption peak dropped from 3299 cm^−1^ to the lower wavenumber of 3294 cm1 when the WPI/PC compounds were produced, widening the bandwidth throughout the range of 3700–3200 cm^−1^. The shift of the absorption peak to a low wavenumber further revealed that WPI and PCs interacted via hydrogen bonding. The amide A band may be brought on by hydrogen bonding, free -OH stretching vibration, or NH bond stretching vibration, according to earlier study findings [34]. At the same time, with the addition of PCs, the movement of the amide I and II bands and the expansion of the characteristic absorption peaks indicated the existence of hydrogen bonds. The similarity of the FTIR spectrum of WPI-Lac/PC compounds to that of WPI/PCs indicated that the interaction between proteins and PCs occurred via a noncovalent bond, based on previous laboratory research [21].

### 3.6. Surface Hydrophobicity

The H_0_ value reflects the number of exposed hydrophobic groups on the protein surface, as well as their structural and functional characteristics. Figure 5 shows the H_0_ values of the protein/PC compounds. The H_0_ content intensity of WPI/PCs and WPI-Lac/PCs (500) was significantly 10 times lower than that of WPI/PCs and WPI-Lac (about 5000) (*p* < 0.05). This shows that in WPI and WPI-Lac, due to the noncovalent bonding between PCs and protein, the binding of PCs to the hydrophobic point on the protein surface was inhibited.

### 3.7. Interfacial Tension

The suspension method was used to determine the dynamic interfacial tension of WPI, WPI/PCs, WPI-Lac, and WPI-Lac/PCs [36]. The oil–water interfacial tension between WPI, WPI/PCs, WPI-Lac, and WPI-Lac/PCs is depicted in Figure 6a for various time intervals. As can be seen from Figure 6, the adsorption process can be divided into two stages [37]: The first stage refers to the rapid absorption of substances such as WPI at the oil–water interface, and the interfacial tension drops sharply. The second part involves a slow reduction in interfacial tension, in which case the protein is more adsorbed at the oil–water interface, forming a relatively stable protein adsorption membrane.

Comparing WPI and WPI-Lac, along with their combination with proanthocyanidins, Figure 6b illustrates that WPI could not easily form a stable droplet morphology. After the protein was combined with proanthocyanidins, the interfacial tension of WPI and WPI-Lac increased; however, compared with WPI/PCs, the interfacial tension of WPI/PCs was smaller. Furthermore, an interfacial protein film formed rapidly after the formation of compounds with PCs, and the oil–water interfacial tension stabilized quickly. This is also consistent with the finding of Tao et al. [37] that WPI composite C3G is more conducive to the formation and rapid stability of interface films.

### 3.8. DSC Analysis

As can be seen from Table 4, the thermal denaturation temperature of WPI-Lac is higher than that of WPI; in Figure 7, the initial annealing temperature and thermal denaturation temperature of WPI-Lac/PCs are significantly higher than those of WPI. This indicates that when the protein is covalently bound to lactose, the thermal stability increases, and after compounding with PCs, its thermal stability is further improved. This may be the result of the Maillard reaction and is consistent with the results of Liu et al. [11]. The complex of PCs creates steric hindrance on the protein surface, inhibiting the stretching of protein peptide chains [11].

### 3.9. Morphology of WPI/PC and WPI-Lac/PC Compounds

Figure 8 depicts the microscopic morphology of WPI/PCs and WPI-Lac/PCs. Although the surfaces of these compounds seemed spherical, WPI/PC compound surfaces were highly rough, but WPI-Lac/PC composite surfaces were smooth and spherical. WPI/PC and WPI-Lac/PC compounds had particles that were smaller than 500 nm in size. Earlier research has demonstrated that WPI and WPI-Lac do not aggregate [18]. Overall, the compound distribution of WPI/PCs is relatively uniform, and the structure of WPI-Lac/PCs has changed, which is caused by the adsorption of PCs on WPI-Lac [35,38].

### 3.10. Analysis of Antioxidant Performance of Compounds

The ability of WPI/PCs and WPI-Lac/PCs to scavenge ABTS and DPPH radicals is a prerequisite for their antioxidant activity, which was also assessed using the FRAP method. The examination of the capacity of the WPI/PCs and WPI-Lac/PCs to scavenge free ABTS and DPPH radicals is shown in Figure 9a,b. In the absence of PCs, there was no difference between WPI-Lac and WPI in terms of the ability to scavenge free radicals. The WPI/PC and the WPI-Lac/PC complexes’ free radical-scavenging abilities were considerably enhanced with the increase in PCs added. This phenomenon suggests that PCs are the primary factor in the enhancement of the complex’s capacity to scavenge free radicals, whilst the protein has just a modest impact. The measurement results of the overall antioxidant activity as determined by the FRAP method are shown in Figure 9c. Between WPI-Lac and WPI alone, there was no discernible change in the total antioxidant capacity. The total antioxidant capacity of WPI/PC and WPI-Lac/PC complexes steadily increased with the increase in PC concentration. The antioxidant capacities of the compounds were in the following order at the same PC content: WPI-Lac/PCs > WPI/PCs > PCs. The research results showed that the presence of PCs was the main factor influencing the antioxidant activity and that the WPI-Lac/PC system had a stronger total antioxidant capacity than the WPI/PC composite system. This difference may be due to the Maillard reaction formation of compounds that resemble melanin and heterocyclic rings, which gives the complex system a higher antioxidant capacity [39].

### 3.11. Application of Glycosylated protein/PC Compound Stabilized Pickering Solutions

The WPI, WPI/PC, WPI-Lac, and WPI-Lac/PC stable Pickering emulsions were characterized by measuring the average particle size and observation with a confocal laser scanning microscope. Figure 10a reflects the average particle size of WPI/PC and WPI-Lac/PC stable Pickering emulsions. Figure 10b shows the microscopic morphology of the Pickering emulsion as observed using a confocal laser scanning microscope [37]. The prepared emulsions were all spherical and uniformly distributed without obvious aggregation. Column I in the figure is the use of fluorescein isothiocyanate (FITC) to stain the protein part of the sample in the aqueous phase, and FITC could emit green fluorescence at an excitation wavelength of 543 nm. Column II is the use of the Nile Red label as the oil phase. Nile Red could emit red fluorescence at the excitation wavelength of 488 nm. Column III was obtained by superimposing the pictures of the oil and water phases in Columns I and II. The experimental results show that the liquids of the four samples are all oil-in-water. Based on previous research [27], the yellow ring structure is caused by the overlapping effect of the green light of FITC and the red light of Nile Red on the oil–water interface, and it also shows a thicker protein adsorption layer, making the emulsion more stable.

## 4. Conclusions

A glycosylated protein/proanthocyanidin complex was synthesized with WPI, lactose, and proanthocyanidins as raw materials. Protein aggregation can be regulated by controlling the amount of proanthocyanidins added. The effects of the interaction between glycosylated WPI and PCs on the antioxidant activity, thermal stability, and interfacial stability of the protein were investigated. The complex (1:1, *w*/*w*, pH 6.0) formed by glycation WPI and PCs had the best stability, and the particle size decreased to 119.4 ± 39.8 nm. PCs affected the secondary structure of glycosylated WPI, and WPI-Lac/PCs showed the best thermal stability (113.33 °C) and antioxidant properties, indicating that glycosylation and the interaction between glycosylated proteins and PCs could enhance the thermal stability and antioxidant properties of proteins. The glycosylated protein/proanthocyanidin complex was characterized by TEM and CLSM. These findings provide new insights into the effects of glycosylation modifications and the effects of interactions between glycosylated proteins and polyphenols on protein properties.

## Figures and Tables

**Figure 1 foods-12-02153-f001:**
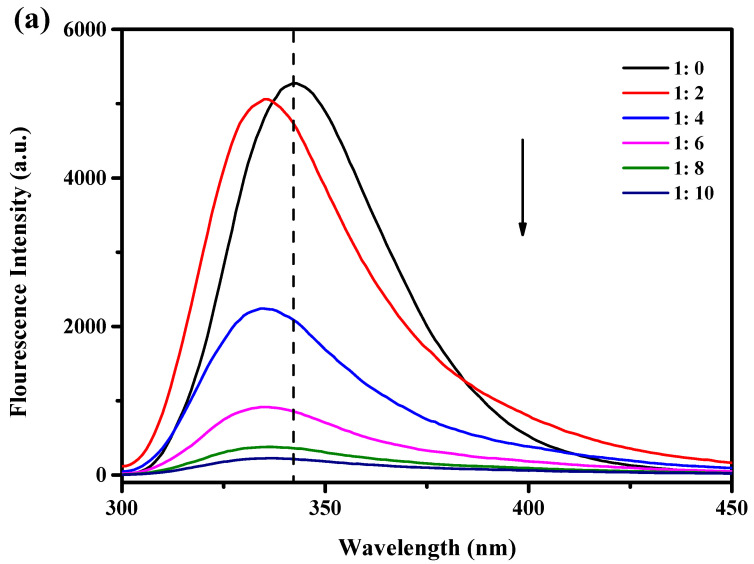
Effect of different concentrations of PCs on the fluorescence intensity of protein: (**a**) WPI/PCs; (**b**) WPI-Lac/PCs, black arrows indicate reduced intensity.

**Figure 2 foods-12-02153-f002:**
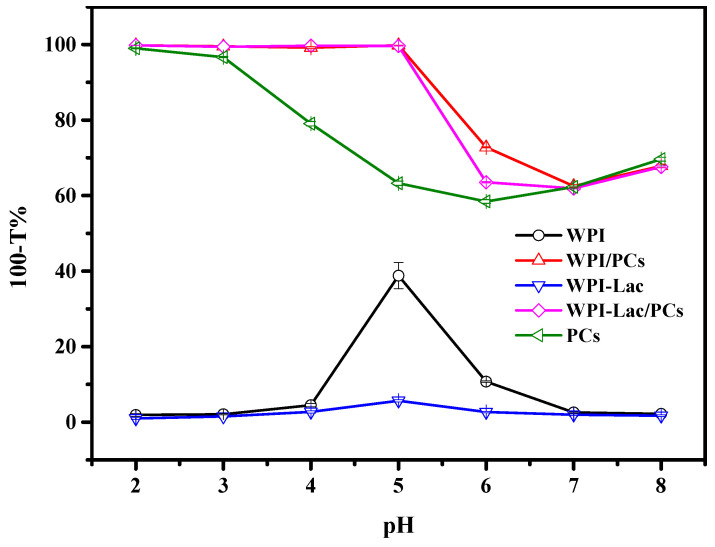
The turbidity (100-T%) of the samples of WPI, WPI/PCs, WPI-Lac, WPI-Lac/PCs, and proanthocyanidins at different pH values.

**Figure 3 foods-12-02153-f003:**
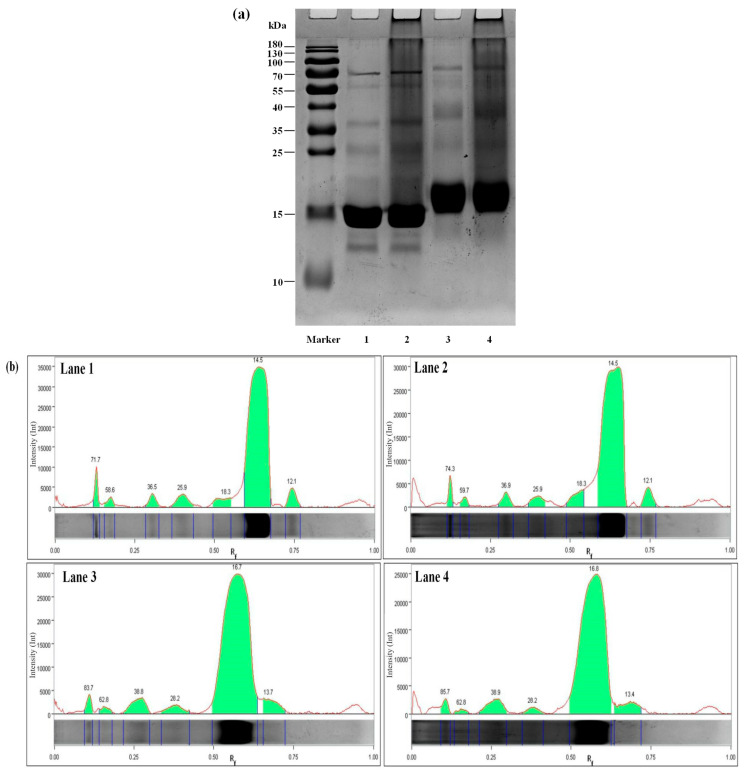
Protein/PC compound SDS-PAGE gel electrophoresis (**a**) and electrophoresis strip image analysis (**b**). Note: Lane Marker, protein marker standard (10–180 kDa); Lane 1, WPI; Lane 2, WPI/PCs; Lane 3, WPI-Lac; Lane 4, WPI-Lac/PCs.

**Figure 4 foods-12-02153-f004:**
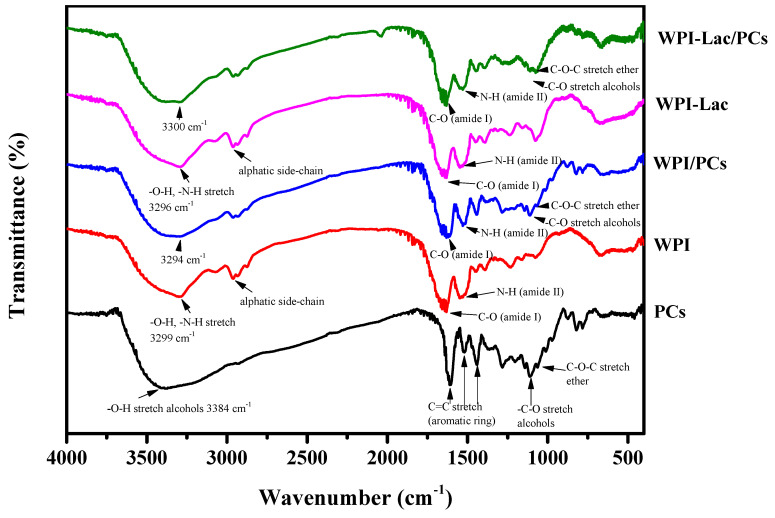
Fourier transform infrared spectrum of protein/PC compounds.

**Figure 5 foods-12-02153-f005:**
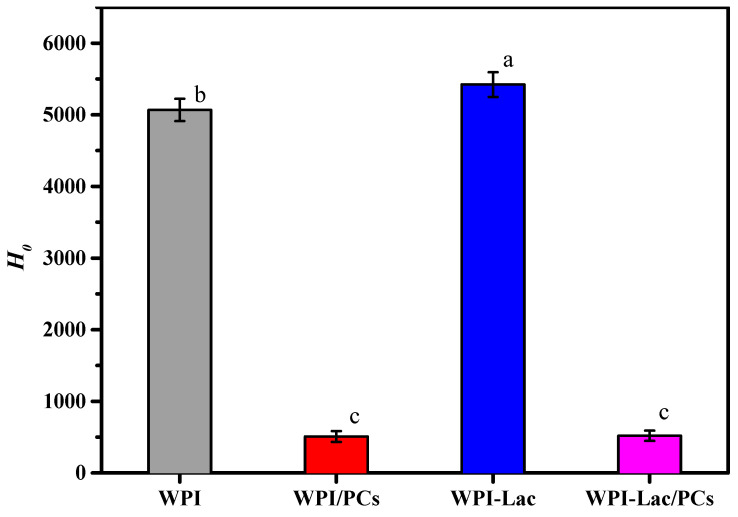
Surface hydrophobicity (*H_0_*) of protein/PC compounds. Different letters indicate a statistically significant difference (*p* < 0.05).

**Figure 6 foods-12-02153-f006:**
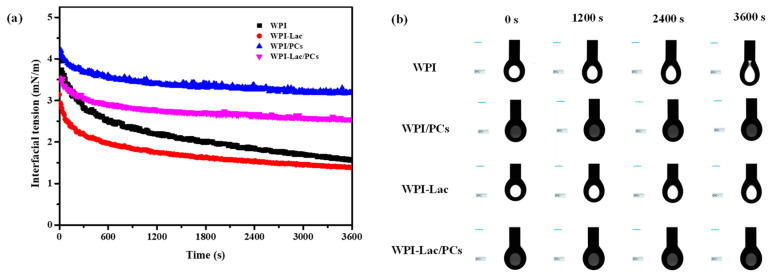
Time–surface pressure evolutions of protein/PC compounds, (**a**) is the interfacial tension (**b**) is the droplet morphology.

**Figure 7 foods-12-02153-f007:**
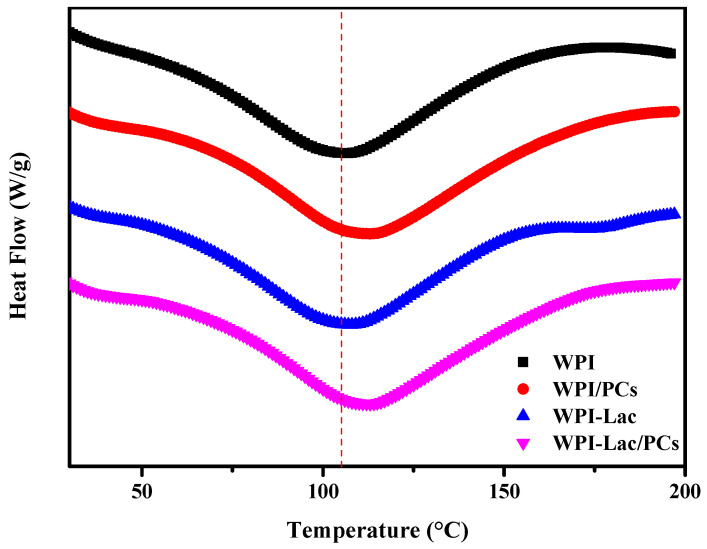
The DSC analysis of protein/PC compounds.

**Figure 8 foods-12-02153-f008:**
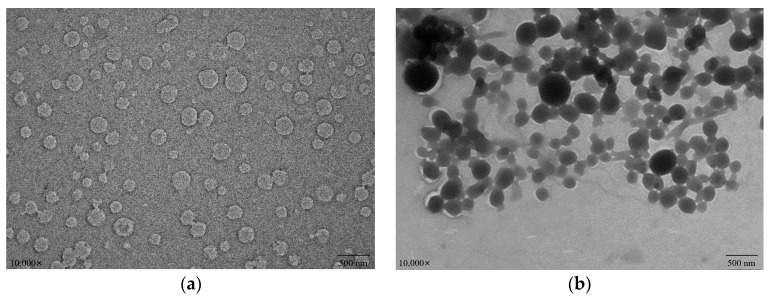
The TEM images of protein/PCs: WPI/PCs (**a**); WPI-Lac/PCs (**b**).

**Figure 9 foods-12-02153-f009:**
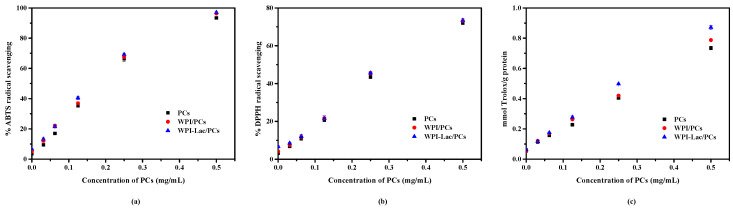
The antioxidation ability of protein/PC compounds: ABTS assay of radical-scavenging activity (**a**); DPPH assay of radical-scavenging activity (**b**); Fe^2+^-reducing power assay of radical-scavenging activity (**c**).

**Figure 10 foods-12-02153-f010:**
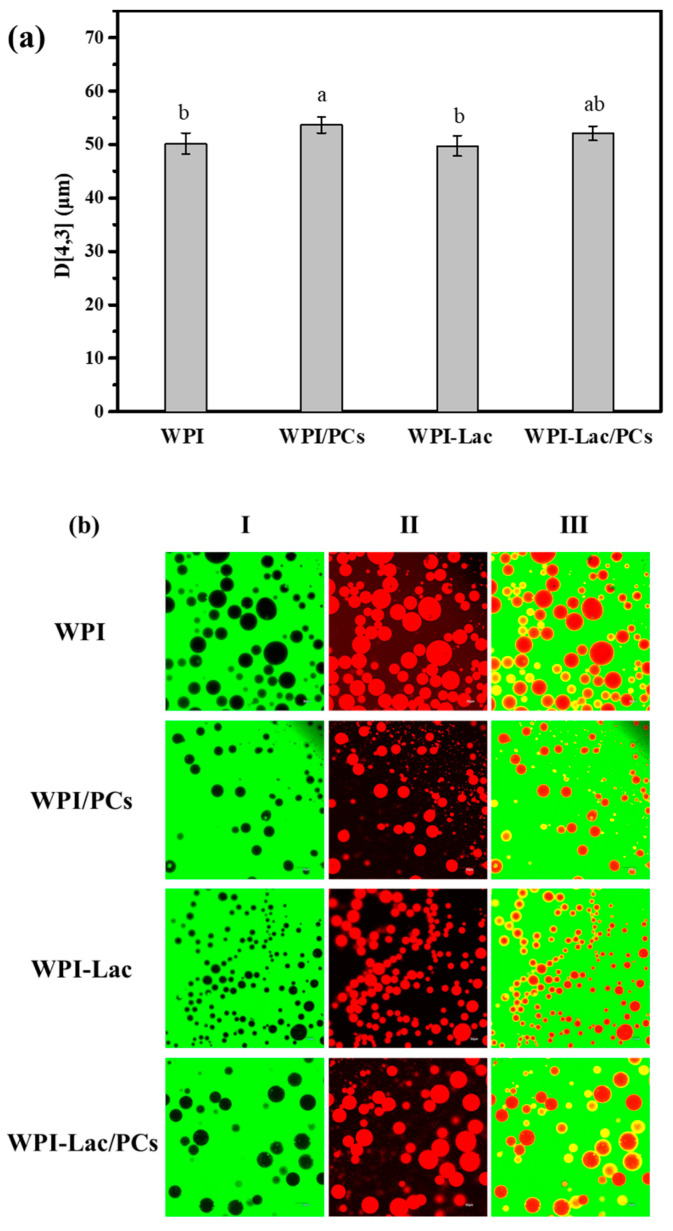
WPI, WPI/PCs, WPI-Lac, and WPI-Lac/PCs stabilize the properties of Pickering emulsion at pH 6.0: particle size (**a**); the CLSM pictures of Pickering emulsion (**b**). Different letters indicate a statistically significant difference (*p* < 0.05).

**Table 1 foods-12-02153-t001:** Particle size, polydispersity, and zeta potential analysis of protein/PC compounds at different mass ratios (pH = 6.0). Different letters indicate a statistically significant difference (*p* < 0.05).

Mass Ratio	WPI/PCs			WPI-Lac/PCs		
	Size (nm)	PDI	Zeta Potential (mV)	Size (nm)	PDI	Zeta Potential (mV)
1:0	246.8 ± 19.52 ^a^	0.522 ± 0.068 ^a^	−23.7 ± 1.1 ^a^	358.8 ± 13.2 ^b^	0.573 ± 0.023 ^a^	−23.0 ± 1.4 ^a^
8:1	209.4 ± 47.8 ^ab^	0.520 ± 0.150 ^a^	−30.0 ± 1.1 ^bc^	444.5 ± 70.6 ^a^	0.470 ± 0.049 ^ab^	−29.0 ± 1.4 ^bc^
4:1	216.1 ± 15.5 ^ab^	0.453 ± 0.063 ^a^	−30.3 ± 0.9 ^bc^	287.8 ± 55.7 ^b^	0.443 ± 0.066 ^b^	−28.9 ± 1.1 ^bc^
2:1	160.8 ± 57.3 ^bc^	0.479 ± 0.135 ^a^	−30.9 ± 0.2 ^c^	197.6 ± 21.6 ^c^	0.334 ± 0.082 ^c^	−30.6 ± 0.4 ^c^
1:1	112.6 ± 10.3 ^c^	0.242 ± 0.039 ^b^	−31.8 ± 1.6 ^c^	119.4 ± 39.8 ^c^	0.226 ± 0.045 ^c^	−30.5 ± 1.4 ^c^
1:2	129.8 ± 28.6 ^c^	0.254 ± 0.023 ^b^	−28.7 ± 0.35 ^b^	126.8 ± 35.9 ^c^	0.251 ± 0.075 ^c^	−28.1 ± 0.6 ^b^

**Table 2 foods-12-02153-t002:** Particle size, polydispersity, and ζ-potential of protein/PC compounds at different pH values. Different letters indicate a statistically significant difference (*p* < 0.05).

pH	WPI/PCs			WPI-Lac/PCs		
	Size (nm)	PDI	Zeta Potential (mV)	Size (nm)	PDI	Zeta Potential (mV)
2	ND	ND	20.8 ± 1.7 ^c^	ND	ND	15.0 ± 1.6 ^c^
3	147.5 ± 4.5 ^b^	0.542 ± 0.015 ^a^	38.5 ± 1.7 ^a^	146.5 ± 3.0 ^cd^	0.623 ± 0.078 ^a^	37.2 ± 1.0 ^a^
4	425.6 ± 79.3 ^a^	0.471 ± 0.055 ^ab^	24.5 ± 0.3 ^b^	200.0 ± 20.3 ^bc^	0.272 ± 0.006 ^c^	22.0 ± 1.0 ^b^
5	411.6 ± 15.5 ^a^	0.464 ± 0.085 ^ab^	−20.2 ± 0.7 ^d^	361.2 ± 5.4 ^a^	0.330 ± 0.019 ^c^	−22.0 ± 0.5 ^d^
6	116.5 ± 29.2 ^b^	0.290 ± 0.019 ^c^	−30.6 ± 0.4 ^e^	117.7 ± 34.5 ^d^	0.309 ± 0.020 ^c^	−33.0 ± 1.0 ^f^
7	319.4 ± 97.3 ^a^	0.376 ± 0.101 ^bc^	−33.8 ± 1.5 ^f^	249.9 ± 46.7 ^b^	0.447 ± 0.094 ^b^	−32.7 ± 1.0 ^f^
8	329.2 ± 59.4 ^a^	0.493 ± 0.042 ^ab^	−30.9 ± 0.8 ^e^	249.9 ± 56.1 ^b^	0.442 ± 0.069 ^b^	−28.0 ± 0.8 ^e^

**Table 3 foods-12-02153-t003:** Electrophoresis strip analysis table.

Band No.	Molecular Weight (kDa)			
	WPI	WPI/PCs	WPI-Lac	WPI-Lac/PCs
1	71.7	74.3	83.7	85.7
2	58.6	59.7	62.8	62.8
3	36.5	36.9	38.8	38.9
4	25.9	25.9	28.2	28.2
5	18.3	18.3	16.7	16.8
6	14.5	14.5	13.7	13.4
7	12.1	12.1	——	——

**Table 4 foods-12-02153-t004:** The thermal property parameters of protein/PC compounds.

Sample	T_m_ (°C)	T_p_ (°C)	ΔH (J/g)
WPI	56.01	106.00	182.5
WPI/PCs	63.18	111.70	258.9
WPI-Lac	58.85	106.80	171.8
WPI-Lac/PCs	64.46	113.33	242.9

## Data Availability

The data presented in this study are available on request from the corresponding author.

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
