# Peer review of "Preparation, Characterization and Antioxidant Activity of Glycosylated Whey Protein Isolate/Proanthocyanidin Compounds"

_foods, 2023, doi:10.3390/foods12112153_

Round 1
Reviewer 1 Report
The authors should improve following section:
2.1. Provide exact compositionof the used WPI and PC
2.2. Provide information on how the pH was adjusted - by which agents
2.5. Describe how did you prepare the samples, did you dillute those with water? Which pH conditions exactly were applied and which agents were used to adjust pH?
2.7. Sample preparations needs to be described more clearly. The mentioned mass ratio 1:100 referrs to what? What was the ratio of potassium bromide with proteins during mixing?
2.12. (1) and (2) what did you use as control for measuring Acontrol?
(3) How did you calculate antioxidant capacity? Describe the procedure for standard curve plotting.
Author Response
Thank you very much for the reviewer's comments and suggestions on this paper. We make the following modifications and explanations.
2.1. Provide exact compositionof the used WPI and PC
1.The protein WPI #9420 was kindly provided by Hilmar Ingredients (Hilmar, CA, USA), the ingredients contain α-whey protein, β-lactoglobulin, bovine serum protein (BSA) and immunoglobulin, mainly β-lactoglobulin.
2.PCs were purchased from Shanghai Yuanye Biotechnology Co., Ltd (Shanghai, China) with the formula C30H26O13.
2.2. Provide information on how the pH was adjusted - by which agents
In this paper, 2 M HCl and 2 M NaOH are used to adjust pH if there is no special explanation.
2.5. Describe how did you prepare the samples, did you dillute those with water? Which pH conditions exactly were applied and which agents were used to adjust pH?
The product obtained in 2.2, that is at a protein concentration of 5 mg/mL, the ratio of protein to proanthocyanidins is 1:1 (w/w), the pH value of 2 ~ 8 was adjusted by 2M HCl and 2M NaOH. The transmittance T(%) of the compound was determined by ultraviolet spectrophotometer (UV-1000, AOE Instrument Co., LTD., China) at 600 nm. Turbidity is expressed as 100-T (%), with ultrapure water as a blank.
2.7. Sample preparations needs to be described more clearly. The mentioned mass ratio 1:100 referrs to what? What was the ratio of potassium bromide with proteins during mixing?
A Fourier infrared spectrometer (NEXUS 670, Nicolet Co., Ltd., USA) was used to analyze the structural changes in the samples. First, lyophilized samples were ground and uniformly mixed with KBr at a weight ratio of 1: 100 and then ground with potas-sium bromide to prepare potassium bromide tablets for analysis with the following parameter settings: wavenumber range of 4000−400 cm−1, resolution of 8 cm−1, and scan times of 16.
2.12. (1) and (2) what did you use as control for measuring Acontrol?
(3) How did you calculate antioxidant capacity? Describe the procedure for standard curve plotting.
Acontrol represents the absorbance of the blank control, and absolute ethanol was used as the blank control in this experiment; Asample represents the absorbance of the sample.
The reducing power of the sample was determined using a oxidation resistance test kit(Beyotime Biotechnology). The working solution was freshly prepared by mixing the dilution solution, TPTZ, and assay buffer solution in a ratio of 10:1:1 (v/v/v). Mix 180 μl of FRAP working solution with 5 μl of diluted sample and incubate at 37°C for 5 min. Determination of Absorbance at 593 nm, it was prepared with FeSO4•7H2O pro-vided in the kit the solution obtains a standard curve.
Reviewer 2 Report
Dear Authors,
Your manuscript presented various analysis methods which whould revealed the in-depth results of the complex. However, you did not explain your results concisely and no discission detected. Please rewrite results to show the merit of your study before resubmission again.
Regards,
Reviewer
Author Response
Thanks for the reviewer's comments, we have made some modifications to the article. Please refer to the attachment for details.

Reviewer 3 Report
Authors prepared glycosylated whey isolate/procyanidin complex. It was characterized using endogenous fluorescence spectroscopy, polyacrylamide gel electrophoresis, Fourier infrared spectroscopy, oil–water interfacial tension, and transmission electron microscopy. The degree of protein aggregation could be regulated by controlling the added amount of procyanidin. Hydrogen bonds and hydrophobic interactions were responsible for interactions between glycosylated protein and procyanidin. The resulting complex had a particle size of about 16 nm. The Authors observed excellent antioxidant and free radical scavenging abilities of obtained complex. It was used also to stabilize a Pickering emulsion as an example of functional properties.
The article contains novel scientific informations and the laboratory work was done very well. Used methods are proper for this kind of research. The level of English language is acceptable. The article should be published after minor corrections mentioned bellow.
Very important paper is not quoted:
Tang C, Tan B, Sun X. Elucidation of Interaction between Whey Proteins and Proanthocyanidins and Its Protective Effects on Proanthocyanidins during In-Vitro Digestion and Storage. Molecules. 2021 Sep 8;26(18):5468. doi: 10.3390/molecules26185468.
In this work, the Authors characterized the interactions between β-Lactoglobulin (β-LG) and α-lactalbumin (α-LA) and oligomeric proanthocyanidins, including A1, A2, B1, B2, B3, and C1, using multi-spectroscopic and molecular docking methods. Fluorescence spectroscopic data revealed that all of the oligomeric proanthocyanidins quenched the intrinsic fluorescence of β-LG or α-LA by binding-related fluorescence quenching. Among the six oligomeric proanthocyanidins, A1 showed the strongest affinity for β-LG (Ka = 2.951 (±0.447) × 104 L∙mol−1) and α-LA (Ka = 1.472 (±0.236) × 105 L∙mol−1) at 297 K. β-LG/α-LA and proanthocyanidins can spontaneously form complexes, which are mainly induced by hydrophobic interactions, hydrogen bonds, and van der Waals forces. Fourier-transform infrared spectroscopy (FTIR) and circular dichroism spectroscopy showed that the secondary structures of the proteins were rearranged after binding to oligomeric proanthocyanidins. During in vitro gastrointestinal digestion, the recovery rate of A1 and A2 increased with the addition of WPI by 11.90% and 38.43%, respectively. The addition of WPI (molar ratio of 1:1) increased the retention rate of proanthocyanidins A1, A2, B1, B2, B3, and C1 during storage at room temperature by 14.01%, 23.14%, 30.09%, 62.67%, 47.92%, and 60.56%, respectively.
There is an error in literature quotation:
441 ]Li Z, Zheng Y, Sun Q, Wang J, Zheng B, Guo Z. Structural characteristics and emulsifying properties of myofibrillar protein-dextran conjugates induced by ultrasound Maillard reaction [J]. Ultrasonics Sonochemistry, 2021, 72: 105458.
Author Response
Thanks for the reviewer's comments, we have made some modifications to the article, please refer to the attachment for details.

Round 2
Reviewer 2 Report
Dear Authors,
Thank you for your responses to all comments. This manuscript is ready to publish at this stage.
Kind Regards,
Reviewer
Author Response
Dear reviewer;
We have modified the language of the article according to the questions of your raised, and the final result was shown in the attachment. We hope the modification could meet your requirements. We will very grateful for your valuable advice.
